# Oriented Insertion of ESR-Containing Hybrid Proteins in Proteoliposomes

**DOI:** 10.3390/ijms24087369

**Published:** 2023-04-17

**Authors:** Lada E. Petrovskaya, Evgeniy P. Lukashev, Mahir D. Mamedov, Elena A. Kryukova, Sergei P. Balashov, Dmitry A. Dolgikh, Andrei B. Rubin, Mikhail P. Kirpichnikov, Sergey A. Siletsky

**Affiliations:** 1Shemyakin & Ovchinnikov Institute of Bioorganic Chemistry, Russian Academy of Sciences, Ul. Miklukho-Maklaya, 16/10, 117997 Moscow, Russia; 2Department of Biology, Lomonosov Moscow State University, Leninskie Gory, 1, 119234 Moscow, Russia; 3Belozersky Institute of Physical-Chemical Biology, Lomonosov Moscow State University, 119991 Moscow, Russia; 4Department of Physiology and Biophysics, University of California, Irvine, CA 92697, USA

**Keywords:** retinal protein, fusion protein, proteoliposomes, proton pump, photocycle, photoelectric potential generation

## Abstract

Microbial rhodopsins comprise a diverse family of retinal-containing membrane proteins that convert absorbed light energy to transmembrane ion transport or sensory signals. Incorporation of these proteins in proteoliposomes allows their properties to be studied in a native-like environment; however, unidirectional protein orientation in the artificial membranes is rarely observed. We aimed to obtain proteoliposomes with unidirectional orientation using a proton-pumping retinal protein from *Exiguobacterium sibiricum*, ESR, as a model. Three ESR hybrids with soluble protein domains (mCherry or thioredoxin at the C-terminus and Caf1M chaperone at the N-terminus) were obtained and characterized. The photocycle of the hybrid proteins incorporated in proteoliposomes demonstrated a higher pK_a_ of the M state accumulation compared to that of the wild-type ESR. Large negative electrogenic phases and an increase in the relative amplitude of kinetic components in the microsecond time range in the kinetics of membrane potential generation of ESR-Cherry and ESR-Trx indicate a decrease in the efficiency of transmembrane proton transport. On the contrary, Caf-ESR demonstrates a native-like kinetics of membrane potential generation and the corresponding electrogenic stages. Our experiments show that the hybrid with Caf1M promotes the unidirectional orientation of ESR in proteoliposomes.

## 1. Introduction

Microbial rhodopsins belong to an important class of membrane proteins that perform the transformation of absorbed light energy to ion transport and sensory functions [1,2,3]. Bacteriorhodopsin (BR) from *Halobacterium salinarum* and green-absorbing proteorhodopsin (PR) from uncultured marine bacterium are the most extensively studied representatives of this family [4,5]. The development of metagenomic approaches enabled the discovery of multiple new retinal proteins, which become the objects of extensive functional and structural studies [3,6,7].

To perform profound characterization of a membrane protein, researchers need to transfer it into a membrane-like environment that could be represented by detergent micelles, bicelles, liposomes, or nanodiscs [8,9,10,11]. In contrast to micelles, liposomes provide the closed circular lipid bilayer and, therefore, enable to study the transport properties of retinal proteins. As was demonstrated by numerous studies, the functional characteristics of microbial rhodopsins in proteoliposomes significantly depend on the lipid characteristics (their charge, length, etc.) and protein chain orientation [12,13]. For example, BR and PR possess similar orientation in the cells with their N-terminus oriented towards the medium and C-terminus to the cytoplasm. However, in the proteoliposomes, these proteins acquire opposite orientation; in addition, as a result, BR mainly pumps protons into the lumen [14] and GPR delivers them to the external medium [15].

Microbial rhodopsins can find various biotechnological applications, including optogenetics [2,16], biooptoelectronics [17], and others. In particular, rhodopsin-containing “protocells” can serve as artificial energy generators. Proteopolymersomes containing BR and F0F1-ATP synthase were capable to perform light-dependent ATP synthesis [18]. The incorporation of both PR and plant-derived photosystem II with ATP synthase into an artificial organelle enabled PMF generation and controllable ATP synthesis, which was utilized for carbon fixation and actin polymerization [19].

Ideally, for the structural and functional studies, protein molecules should be uniformly oriented in a lipid bilayer [20,21]. In the native cells, unidirectional protein orientation in the membrane is determined during the complex folding and insertion process with the participation of chaperones, components of the translocation complex, and other factors [22,23,24]. However, in the proteoliposomes, such a situation is rarely observed. In most cases, a mixture of orientations occurs resulting in a decreased functional signal, in particular, in impaired ion transport properties that can complicate characterization and applications of the microbial rhodopsins [20,25]. To ensure the uniform orientation of membrane proteins in proteoliposomes, several technologies were proposed. The binding of the PR C-terminal His-tag to Ni-NTA-coated silicate beads allowed the insertion of the protein into lipid vesicles with C_out_ orientation [26]. Tunuguntla et al. demonstrated that the surface charge of the lipid bilayer can assist the asymmetrical insertion of a retinal protein, leading to a preferential proton movement in a corresponding direction [27].

In the cells, large N- and C-terminal domains can drive the oriented insertion of the membrane proteins [28,29]. The ability of protein engineering to add artificial soluble modules to the target membrane proteins was successfully exploited by Ritzmann et al. [30] to mimic this natural approach. Fluorescent proteins mCherry and GFP were genetically attached to N- and C-termini of the PR, correspondingly. That guided PR insertion into proteoliposomes with their free termini pointed inside the lumen. The obtained fusion proteins performed light-driven proton transport in the opposite directions according to the results of the pH changes measurements.

In the current study, we have applied this approach to obtain proteoliposomes with the uniform orientation of the proton-pumping retinal protein from *Exiguobacterium sibiricum*, ESR [31]. This microbial rhodopsin belongs to the proteorhodopsin family [32,33]; however, it contains an unusual lysine residue that acts as a proton donor to the Schiff base [34]. Previously, we have shown that ESR and its mutants can be successfully incorporated into the proteoliposomes that produce a light-induced electrogenic signal upon the association with a phospholipid-impregnated collodion film [35,36,37,38]. Analysis of the electrogenic response kinetics indicated that a small fraction of the protein has an opposite orientation in the bilayer; thus, complicating data interpretation. We have produced ESR fusions with three soluble protein partners (mCherry or thioredoxin at the C-terminus, and Caf1M chaperone at the N-terminus), and demonstrated that they differ in providing oriented insertion of the retinal protein into the lipid bilayer and maintaining vectorality of proton transport in a proteoliposome. Our experiments show that the N-terminal position of the fusion partner is critical for the uniform orientation of the ESR in proteoliposomes.

## 2. Results

### 2.1. Recombinant Fusion Protein Construction and Expression

The amino acid sequence and three-dimensional structure of ESR are homologous to those of PR; moreover, similar to PR, ESR has a negatively charged N-terminus and a positively charged C-terminus. The direction of proton transport in proteoliposomes is similar for both proteins, at least at neutral pH values. Therefore, we decided to exploit the utility of the fusion protein approach developed for PR [30] to ensure the uniform orientation of ESR in the bilayer. In this work, we have used three highly soluble proteins as fusion partners. As C-terminal fusions, a fluorescent protein mCherry and *E. coli* thioredoxin (Trx) were used (Figure 1). The molecular weight of mCherry is more than two times larger than that of Trx (Table 1); as a result, the effect of the fusion partner size could be also assessed.

To obtain an N-terminal fusion with a membrane protein with a N_out_-C_in_ configuration, the fusion partner should be translocated to the periplasmic space. Previously, a hybrid of PR with mCherry was constructed using the Skp signal sequence, which provided Sec-dependent translocation of the N-terminal soluble domain [30]. However, the yield of the obtained protein mCherry-PR was relatively low (about 0.5 mg/g of cell pellet). In our case, expression of the hybrid protein mCherry-ESR was not detected in *E. coli* cell membranes with the use of protein electrophoresis and Western blot with anti-His antibodies (Figure 1A,B, lane 4). We presumed that fusion with a secreted bacterial protein would be beneficial to increase the synthesis level of the hybrid protein. With this aim, we selected a highly soluble protein Caf1M from *Y. pestis*, which belongs to the molecular chaperone superfamily and is abundantly expressed in *E. coli* periplasm [41]. The gene coding for the Caf1M precursor with its own signal sequence was cloned in frame with the ESR gene, resulting in N-terminal fusion with the target protein (Figure 1C). In all constructs, the fusion partners were separated by the flexible linker (GSGSGGGGS).

Fusion proteins were expressed under the control of the T7lac promoter as previously established for the wild-type ESR [31]. Similarly to the wild-type ESR, the hybrids were detected in the fraction of the induced C41(DE3) cells obtained after high-speed centrifugation of the lysate that confirmed their successful incorporation into the membrane (Figure 1A,B). The hybrid proteins were solubilized in DDM micelles and purified by Ni-affinity chromatography due to the presence of the C-terminal hexahistidine tag. The obtained yield of the purified proteins was 2.4 mg for ESR-Cherry, 3.2 mg for ESR-Trx, and 1.2 mg for Caf-ESR from 1 g of wet biomass.

### 2.2. Characterization of the Fusion Proteins in Micelles

Purified fusion proteins in the micelles of non-ionic detergent DDM were characterized by absorption spectroscopy and studies of flash-induced kinetics of absorption changes at selected wavelengths (flash photolysis). At pH 7.0, the absorption maxima of the ESR-Trx and Caf-ESR hybrids were identical to that of the wild-type ESR (528 nm, Figure 1D). In the spectrum of the ESR-Cherry hybrid, two characteristic maxima were observed, a major peak at 541 nm and a smaller one at 587 nm, presumably resulting from a convolution of the ESR and mCherry curves.

Previously, we demonstrated that ESR undergoes a photocycle that includes several intermediates with different absorption maxima: ESR_532_→K→L_530_→M1_400_↔M2_400_↔N1_580_↔N2/O_540_→ESR [31,32,42]. The kinetics of light-induced absorption changes in the wild-type ESR and the hybrid proteins were examined at four characteristic wavelengths, 410, 510, 550, and 590 nm (Figure 2). In all proteins, the decay of the K intermediate and formation of the L intermediate occur on the microsecond time scale, and are accompanied by the decrease of absorption at 590 nm, 550 nm, and 510 nm. At pH 7.0, almost no M intermediate was observed in the photocycle of all studied proteins as was observed earlier for the wild-type ESR [31].

It was shown that the pKa of M accumulation in ESR depends on the environment. It is lower in lipids and lipid-like detergents than in DDM, which results in large M accumulation at neutral pH [34]. At pH 9.0, the formation of the M intermediate is reflected in an increase in absorption at 410 nm and contains three components with τ_1_ ~4–9 μs, τ_2_ ~80–180 μs, and τ3 ~2–3 ms (Table 2). Noteworthy, the contribution of the slowest millisecond component of M rise increases in the hybrid proteins up to 56% in ESR-Trx and Caf-ESR, and to 75% in ESR-Cherry in comparison with 33% in the wild-type ESR. The decay of M involves two components, 10–12 ms (~70%) and 293–322 ms (~30%). It results in the production of red-shifted intermediates that decay with a time constant of 60–83 ms.

Overall, the photocycle characteristics of the obtained fusion proteins in DDM micelles were similar to those of the wild-type ESR. To assess the efficiency of the hybrid protein incorporation into the lipid bilayer, their orientation, and functionality, we studied the kinetics of light-induced changes of transmembrane potential difference (ΔΨ) for the proteoliposomes with the hybrid proteins attached to a collodion film along with the light-induced absorption changes at selected wavelengths, as described below.

### 2.3. Photocycle of the Hybrid Proteins in Proteoliposomes

During the first 10 μs of the photocycle, the decay of a red-shifted K-like intermediate is accompanied by the decrease of absorption at 590 nm in the wild-type ESR and the fusion proteins with τ ~3–5 μs at pH 7.5 (Figure 3). In the proteoliposomes reconstituted with the wild-type ESR, the formation of M was clearly observed as an increase of absorption at 410 nm with time constants ~3, 14, and ~82 μs (Appendix A). Contrary to that, the amount of M was very small in all hybrid proteins at neutral pH (Figure 3), indicating that the pKa of M accumulation is increased in hybrids, presumably as the result of a higher pK_a_ of the Schiff base in M.

It should be noted that only a small fraction of K converts fast to M in the hybrid proteins incorporated in proteoliposomes at neutral pH. The larger part decays much slower with time constants in the order of 10–20 ms (Figure 3).

The amplitude of the absorption changes at 410 nm was the largest in the ESR-Trx hybrid, where the formation of the M intermediate proceeds via two kinetic components with time constants ~5 and ~130 μs (Appendix A). Similar to the wt, the fast phase of M decay with τ ~0.4 ms and the slow phase with τ ~4 ms were observed in this hybrid. In both proteins, a decrease of absorption at 410 nm at this timescale is accompanied by an increase at 510, 550, and 590 nm, indicating accumulation of N1 (maximal changes at 550 nm) and N2/O states (maximal changes at 590 nm) [32]. The decay of the N2/O intermediates and return to the initial state in the hybrid proteins proceeds in one kinetic component with τ ~20–23 ms (Appendix A). This value is close to the wild type (21.6 ms, Appendix A).

In the ESR-Cherry, corresponding absorption changes at 410 nm are about four times smaller than in ESR-Trx and include kinetic components with time constants ~4 and ~90 μs for M formation, and 3.4 ms for M decay (Appendix A). Return to the ESR state occurs with τ ~23 ms. An increase of the pH in the proteoliposome suspensions containing ESR-Cherry to 8.5 resulted in the rise of the M intermediate with τ_1_ ~3 μs and τ_2_ ~58 μs (Figure 4A). At this pH value, the decay of M to the N2/O state proceeds with two time constants ~3.3 and ~12.5 ms, and the recovery of the initial state occurs with τ ~70 ms. At pH 9.5 (Figure 4B), the amount of the M intermediate was larger, while the whole photocycle became slower (the slowest time constant ~262 ms).

In general, the photocycle of Caf-ESR at pH 7.5 is similar to those of other hybrid proteins (Appendix A). The fast kinetic components with τ ~3 and 14 μs coupled to the formation of the M state in the wild-type ESR were not detected in Caf-ESR. M rise proceeds with a single time constant of 54 μs and M decay has τ ~3 ms. Remarkably, the photocycle transitions in Caf-ESR demonstrate better consistency with the kinetic components of the electrogenic response (see below). At the same time, it should be noted that direct electrometry provides a much better signal-to-noise ratio; that is, it is more sensitive compared to spectral measurements.

It is worth mentioning that the kinetics of light-induced absorption changes of the hybrid proteins in proteoliposomes at pH 7.5 resemble those of the DDM-solubilized wild-type ESR measured at the same pH value [32]. In that conditions, the predominance of the long-lived red-shifted K-like species measured at 590 nm and a negligible amount of M were the characteristic features of the photocycle. It was attributed to the influence of the protonated state of His57, closely interacting with the proton acceptor from the Schiff base, Asp85, on its p*K*a in M and accumulation of the latter [32].

### 2.4. Electrogenic Response of the Fusion Proteins Incorporated into Proteoliposomes

Studies of the kinetics of membrane potential generation are of great value for qualitative and, in some cases, quantitative assessment of the efficiency of charge translocation by proton pumps and their orientation in proteoliposomes. For example, a decrease in the total relative amplitudes of millisecond phases points to the reduced efficiency of proton pumping due to reverse reactions [36]. At the same time, the rapid discharge of liposomes indicates an increase in the permeability of liposomes to protons. A decrease in the total amplitude of the photoelectric response and the appearance of negative electrogenic components indicate the misorientation of proteins in proteoliposomes.

The kinetics of light-induced changes of transmembrane potential difference (ΔΨ) for ESR-containing proteoliposomes attached to the phospholipid-impregnated collodion film was examined for the wild type and the hybrid proteins along with the light-induced absorption changes of proteoliposomes at selected wavelengths. Figure 5 demonstrates the kinetics of ΔΨ generation in the wild-type ESR, and the hybrid proteins at pH 7.5 and 6.5 on a piecewise linear a and logarithmic time scales. The curves with marked positive and negative phases of the response are provided in the Appendix A. The corresponding absorption changes at pH 7.5 are shown in Figure 3.

After the flash, all three hybrid proteins produced a photoelectric response, which corresponded to the proton transfer from the inside of the liposomes to the bulk, similar to the wild-type ESR [35]. However, the amplitude of ΔΨ generated by the proteoliposomes with ESR-Cherry and ESR-Trx was on average about two times smaller than that of the proteoliposomes with Caf-ESR under the same conditions (Figure 5). Moreover, the number and sign of the components and the rate constants of the electrogenic response exhibited substantial differences between these proteins as discussed below.

#### 2.4.1. ESR-Cherry

Due to the instability of the proteoliposomes containing ESR-Cherry at pH 6.5, the photoelectric response of this hybrid was measured only at pH 7.5. The maximum amplitude is more than two-fold lower in this hybrid than in the wild type and the relative contribution of phases is strongly altered. In response to the flash, the microsecond components of ΔΨ generation are resolved, which correspond to the electrogenic proton transfer through the proteoliposome membrane (~2.7 and ~79 μs, Appendix A and Figure 5). Similarly to the wild-type ESR, these phases are associated with deprotonation of the Schiff base (M-state formation). The overall contribution of these components comprises 11.3% of the total amplitude, which is ~2.3 times greater than in the wild type (5%).

The relative increase of the amplitude of the electrogenic phase associated with proton transfer from the Schiff base to the primary acceptor Asp85 and the formation of the M state could be explained assuming a decreased efficiency of transmembrane proton transport by the hybrid protein incorporated in proteoliposomes in comparison with the wild-type ESR in the stages of photocycle that involve M decay. In that case, the observed effect is a consequence of a decrease in the amplitudes of subsequent electrogenic stages associated with reprotonation of the Schiff base in the catalytic cycle.

Electrogenic components of M decay in ESR-Cherry with τ ~0.4, 1.55, and ~7.2 ms reflect the electrogenic proton transfer reactions in the transitions M→N(O)→ESR similarly to the wild type ESR [35]. The time constants of these components demonstrate significant differences from the corresponding components of the kinetics of light-induced absorption changes (3.4 and 23.3 ms, Appendix A). Remarkably, the direction (sign) of membrane potential generation for two of these components differs from the wild-type ESR. In contrast to the 0.6 ms electrogenic phase in the wild type, the electrogenic phase with τ ~0.4 ms in ESR-Cherry has a negative sign reflecting the decrease in membrane potential generation. In opposite to the electrogenic phase with τ ~0.4 ms, the next electrogenic step with τ ~1.55 ms has the same direction as the microsecond electrogenic components associated with the formation of the M state and produces the main contribution (>88%) to positive electrogenesis in this hybrid. It is close in time to the electrogenic phases (0.6 ms, 3.4 ms) in the wild type that are associated with the M→N(O) transition and presumably corresponds to the Schiff base reprotonation in ESR-Cherry during this transition in the parallel 3.4 ms optical phase.

Similarly to the ~0.4 ms component, the 7 ms electrogenic phase in ESR-Cherry has a negative sign reflecting the decrease of the membrane potential. Corresponding absorption changes presumably are too small and are not resolved. It should be mentioned that the electrogenic phase with τ ~18 ms that in the wild type corresponds to the proton release at the extracellular surface in the N2/O→ESR transition is absent in the case of the hybrid protein. The presence of the negative phases in the kinetics of ΔΨ generation associated with a reverse proton transfer to the Schiff base was revealed previously in the K96A mutant of ESR [36]. In ESR-Cherry, the internal proton donor for the SB Lys96 is present, which should largely prevent reprotonation of the Schiff base from the extracellular side. However, similar reverse reactions could not be totally excluded. The more likely explanation for the 0.4 ms negative phase is a superposition of the kinetics from oppositely oriented ESR molecules in the proteoliposomes, a misorientation.

At first sight, the presence of the protein species with opposite orientations should lead to the decrease of the amplitude of the electrogenic phases, but not to the change in the ratio of the amplitudes of the microsecond and millisecond phases. However, the different environment of the oppositely oriented molecules that could result in altered kinetics should be taken into account. The supplementary explanation assumes that the proteoliposomes with this hybrid are intrinsically leaky, causing the fast passive reverse flow of the charges with ca 7 ms that prevent the accumulation of potential on the millisecond time scale. Poor stability of the samples at pH 6.5 could also be associated with such leakiness and indirectly supports this presumption. Previously, it was shown that the appearance of faster membrane discharge components indicates an increase in the permeability of liposomes for protons due to the addition of uncouplers or channel-promoting agents [43].

#### 2.4.2. ESR-Trx

Similarly to the ESR-Cherry, studies of the electrogenic response of the hybrid protein ESR-Trx at pH 7.5 revealed an increase in the relative amplitude of the electrogenic phases that correspond to the M-state formation in comparison with the wild-type ESR. However, their contribution was lower than that in ESR-Cherry (6.7 vs. 11.3%, Appendix A).

The main contribution to the positive electrogenesis is provided by the electrogenic phase with τ ~1.4 ms, which corresponds to the M→N/O transition in this hybrid. The presence of the negative electrogenic components in the millisecond timescale that correspond to M decay and recovery of the initial state was also revealed in ESR-Trx with time constants that were 2-4 times slower than in ESR-Cherry (0.9 vs. 0.4 and 29 vs. 7 ms). Similarly to the ESR-Cherry, the positive electrogenic phase corresponding to the recovery of the initial state of the ESR and proton release at the extracellular surface of the protein was not detected in this hybrid protein. Instead, the phase with a negative sign is resolved with a similar time constant (29 ms). The relative contribution of this component is about two times smaller than that of the corresponding phase in ESR-Cherry (24 vs. 51%).

At pH 6.5, fast electrogenic components in the microsecond time scale (~4 and 30 μs), which together with slower microsecond electrogenic phases reflect the proton transfer from the Schiff base to Asp85 (formation of the M state) in the wild-type ESR, are not resolved in ESR-Trx (Appendix A). The amplitude of the negative electrogenic phase with τ ~0.9 ms increases significantly, while the contribution of the positive phase with τ ~2.5 ms presumably corresponding to the electrogenic proton transfer from the bulk to the Schiff base is markedly decreased. A very slow positive electrogenic phase (τ ~200 ms) appears after that.

Taken together, the features of the kinetics of ΔΨ generation in ESR-Trx hybrid protein at pH 7.5 and even more so at pH 6.5 point to the considerable distortion of the photoelectric response, which is presumably caused by superposition of the signals from oppositely oriented protein molecules in the proteoliposomes and by reverse reactions of the photocycle. Based on the ratio of the sum of the amplitudes of the negative and positive electrogenic phases, we can estimate that the fraction of misorientation could reach up to ~25%. The appearance of the 200 ms phase is most probably a result of the accelerated passive leak through the membrane of the proteoliposomes containing ESR-Trx.

#### 2.4.3. Caf-ESR

In the kinetics of ΔΨ generation in proteoliposomes containing the Caf-ESR hybrid protein at pH 7.5, the relative amplitude of the microsecond electrogenic components corresponding to the formation of the M state is similar to that in the wild type and even smaller (~3% instead of ~5%, Appendix A, [35]). The same tendency is observed in the comparison of the photoelectric response of the hybrid protein with that of the wild type at pH 6.5 (1.2% instead of 3.2%, Appendix A). In contrast to other hybrid proteins, the relatively low contribution of the microsecond phases in Caf-ESR demonstrates the native-like kinetics of subsequent electrogenic stages and the high efficiency of the proton transport in this protein.

It should be noted that the photoelectric response of the Caf-ESR in the microsecond scale includes a kinetic component with a time constant ~3 μs, which is presumably associated with the decay of the K state coupled with a decrease in absorption at 590 nm. Its relative amplitude is close to that of the wild type (~1.5%). In BR, the process of the L-state formation from the K state corresponds to the charge transfer over a distance of ~4.5 Å [44] that correlates with the amplitude of this component in the wild-type ESR, taking into account that in ESR, the process of formation of K and the transition of K to L possess close characteristic time constants [35]. Therefore, decreased yield of the microsecond electrogenic components in the Caf-ESR is likely explained by the slow formation of the M intermediate as discussed below.

In the microsecond time range, a single kinetic component with the time constant ~60 μs reflects deprotonation of the Schiff base and proton transfer to the acceptor residue Asp85. This constant reasonably corresponds to the ~54 μs component of the photocycle (Appendix A). In the wild-type ESR, two kinetic components in this time range correspond to the formation of M, with τ ~24 and 100 μs with a total amplitude, which is ~2.3 times greater than that of the ~60 μs phase. These data point to the decreased rate of M formation in the hybrid (Figure 5).

Remarkably, the amplitude of the corresponding absorption changes at 410 nm (with τ ~54 μs) in the hybrid is significantly smaller compared to the sum of the amplitudes of the 24 and 100 μs kinetic components in the photocycle of the wild-type ESR (Appendix A). In the wild-type ESR, a decrease in absorption at 590 nm in this time interval is observed, while in the hybrid there is an increase in absorption, which probably corresponds to a small production of the N state. We assume that this component in the Caf-ESR hybrid includes part of the electrogenic process caused by the transition of M to N (reprotonation of the Schiff base). Apparently, the transition between early and late red-shifted intermediates in the Caf-ESR proceeds without accumulation of M (Figure 3D). Since the distance of electrogenic proton transfer during M→N transition is significantly greater than during the formation of M from K/L, a small amount of the emerging intermediate N makes a slightly greater contribution to electrogenesis, explaining the difference between the amplitudes of the kinetics of absorption changes at 410 nm (9 times) and corresponding kinetics of ΔΨ generation (~2.3 times) between the wild type and the Caf-ESR hybrid.

Unlike in the wt, in the Caf-ESR, the M→N/O transition at pH 7.5 is accompanied by a single positive electrogenic phase (τ ~1.6 ms), which reflects reprotonation of the Schiff base. The relative contribution of this electrogenic component (61%) is close to the sum of contributions of two electrogenic phases in the wild type, which are connected with the M→N/O transition (74.5%).

The absence of the negative electrogenic components in the kinetics of ΔΨ generation in Caf-ESR at pH 7.5 distinguish this hybrid protein from the two other fusions, ESR-Cherry, and ESR-Trx, and resembles the kinetics of the wild type ESR at this pH. The negative components are also absent at pH 6.5 in contrast to other variants including the wt ESR, indicating the absence of back reactions and misorientation in this protein.

In contrast to other hybrid proteins, the last electrogenic phase that corresponds to the N2/O→ESR transition (τ ~17.6 ms) in Caf-ESR has a positive direction and contributes about 36% to the overall response. This value corresponds to the proton release from the acceptor site to the bulk phase, similar to the wild type ESR [35].

At pH 6.5, two millisecond electrogenic components (0.29 ms and 4.2 ms) with relative contributions to the full amplitude of 18.7 and 47.3%, respectively, related to transitions M→N1 and M→N1→N2/O are resolved in the kinetics of the photoelectric response of Caf-ESR (as in the wild-type protein). The slowest electrogenic component (with ~17 ms and 32.9% contribution) refers to the transition N2/O→ESR. At both pH values, no accelerated passive discharge of the liposome was observed, indicating that Caf-ESR does not increase the permeability of the liposome membrane for ions unlike the other two hybrids.

## 3. Discussion

Fusion proteins are widely used to provide membrane protein expression in *E. coli* cells for structural and functional studies [45,46,47,48]. Among the most popular fusion partners are glutathione-S-transferase [49], ketosteroid isomerase [50], maltose-binding protein (MBP) [51], and thioredoxin [40,52]. Frequently, the construction of membrane protein fusions is aimed to obtain higher expression levels. In that case, fusion with highly hydrophobic proteins can promote the formation of the inclusion bodies and accumulation of the insoluble protein requiring subsequent refolding step [50]. When membrane localization and the functional state of the targets are the decisive factors, the usage of highly soluble proteins (e.g., MBP, thioredoxin) is desirable. For example, successful recombinant expression of BR in *E. coli* was achieved by the fusion with MBP [53] and Mistic [54]. Fusions of the target polypeptides with GFP and other fluorescent proteins can be also used for seamless monitoring of their synthesis level during expression optimization studies [55]. The ability of the partners to provide proper orientation of a target protein in the artificial membrane that enables more precise control of their functional activity represents a new perspective field of the fusion technology application [30].

In the current work, we employed three highly soluble proteins as fusion partners to provide a directed orientation of ESR in the proteoliposomes. mCherry is a popular monomeric derivative of DsRed from *Discosoma* sp. [39]. Fusions with mCherry are frequently used to monitor target proteins’ biogenesis, localization, folding, and other properties [56]. Thioredoxin from *E. coli* facilitates recombinant protein production in soluble form and increased yield [40]. Both proteins have cytoplasmic localization in bacterial cells and, therefore, were attached to the C-terminal end of ESR. It should be mentioned that mCherry could also be secreted through the Sec pathway and retains fluorescent properties in the periplasm of *E. coli* cells [30,56]. Unfortunately, our attempts to obtain N-terminal Cherry-ESR fusion were unsuccessful; expression of this recombinant protein was not detected in *E. coli* cells. This is in contrast to previously published data where Cherry-PR hybrid protein was obtained; however, with limited yield [30]. Consequently, we decided to use a secreted Caf1M protein with the aim to increase the expression yield of the N-terminal fusion with ESR.

All three hybrids were efficiently expressed in *E. coli* cells, though with different yields. It should be mentioned that the synthesis level of the mature Caf-ESR is limited by the capacity of the cellular machinery that provides protein export to the periplasmic space (Sec-translocon). Not surprisingly, the yield of the Caf-ESR hybrid was the lowest among the three fusion constructions.

Since the photocycle time constants of the hybrids in detergent micelles were close to those of the wild-type ESR, we studied the light-induced absorbance changes and transmembrane potential generation in proteoliposomes reconstituted with these proteins in comparison with ESR-containing liposomes. Interestingly, a small amount of the M intermediate was observed at pH 7.5 only in the photocycle of ESR-Trx incorporated into proteoliposomes, while the recombinant ESR at this pH produced large absorbance changes at a characteristic wavelength 410 nm (Figure 3, [32]). To provide an explanation for this phenomenon, we examined the pH dependence of the photocycle of the ESR-Cherry in a liposome environment and found that the M intermediate in this protein was detected only at pH > 8.5. Similar pH-dependence was observed earlier for the wild-type ESR [32] and in this work for the fusions solubilized in DDM micelles. It was shown previously that pK_a_ values of the photocycle transitions in ESR are very sensitive to the environment. For example, in DDM-solubilized samples, the corresponding values are shifted by 2–2.5 units to the higher pH range [32], while in the micelles of lipid-like detergent LPG, they are more similar to those in liposomes [34]. In other words, the hybrid proteins reconstituted in liposomes demonstrated the photocycle properties (namely, pH dependence of M formation), which are characteristic for the DDM-solubilized samples, which is apparently from the influence of fusion properties on the pKa of the counterion in M and, specifically, the pK_a_ of His57 that closely interacts with proton acceptor Asp85.

It is widely accepted that charged headgroups of the lipid bilayer can affect the proton concentration and surface potential near the interface [57,58]. For example, the addition of anionic lipids (cardiolipin, phosphatidic acid, and others) to POPC-based large unilamellar vesicles resulted in a systemic shift of the pK_a_ of the pHLIP peptide insertion, which depends on electrostatic surface potential [58]. The local properties of such regions could differ from the measured pH in the bulk solution, and these differences presumably stand for observed discrepancies in the pK_a_ values in the micelle and proteoliposome environments. In the proteoliposomes reconstituted with hybrid proteins, the presence of the soluble domains can eliminate/neutralize the effect of the lipids by providing highly hydrophilic space at the membrane exterior. This creates a situation that is close to the one observed in DDM-solubilized samples, resulting in similar pK_a_ values of M formation. In other words, properties of ESR with a soluble domain incorporated into the proteoliposome resemble those of the same protein in a detergent micelle due to the presence of a hydrophobic shell (lipid moiety) and a hydrophilic exterior (soluble domain).

Indirect evidence for the proposed explanation is provided by the fact that among the studied fusion proteins ESR-Trx demonstrated the largest amount of the M intermediate in proteoliposomes at pH 7.5. This could be attributed to the smaller size of the thioredoxin partner (~12 kDa in comparison with ~27 kDa for mCherry and Caf1M), and, correspondingly, to the smaller neutralizing effect of the influence of lipid headgroups (which include negatively charged phosphatidylinositol, zwitterionic phosphatidylethanolamine, and phosphatidylcholine, and other components of azolectin used for proteoliposome preparation).

The contribution of the soluble domain charge to the charge on the liposome surface should be also considered as a possible reason for the altered pK_a_ values of M formation in the hybrid proteins. Placing of a negatively charged domain at the proteoliposome exterior should increase its negative potential and vice versa [27]. However, taking into account that three fusion partners possess different pI values (Trx and mCherry are negatively charged at neutral pH, and Caf1M is slightly positive, Table 1), this explanation is less likely and requires a more detailed study of charge distribution in these proteins.

The decrease of the total amplitude of rapid electrogenic phases coupled to M formation in all hybrid proteins also represents an interesting phenomenon that could be associated with the slow accumulation of the M state in the hybrids. In the photocycle of the hybrid proteins, M formation presumably occurs in a submillisecond time scale, simultaneously with the M decay; therefore, the M state does not accumulate. Correspondently, electrogenicity, which is coupled with this process, is partially shifted towards the millisecond scale of the kinetics of ΔΨ generation. As a result, the relative amplitude of the electrogenic microsecond components is smaller than in the wild type. We can presume that the observed high ratio of the amplitudes of the micro- and millisecond electrogenic components in the ESR-Cherry and ESR-Trx hybrids is even higher due to this effect.

It should be mentioned that microsecond phases make a relatively low contribution to the electrogenesis in the wild-type ESR (about 6%, [35]). Therefore, even large differences in the kinetics of M-state formation are only moderately reflected in the kinetics of ΔΨ generation. Due to this reason, electrogenic responses of the wt and Caf-ESR look similar, especially on the linear time scale, while their photocycles are dramatically different.

In spite of the overall similarity of the photocycle characteristics, the hybrid proteins demonstrated different efficiencies in facilitating directed orientation of the ESR in the lipid bilayer, efficiency, and vectorality of proton transport. This conclusion is supported by the following observations. (1) The amplitudes of the photoelectric response from the proteoliposomes containing Caf-ESR protein were almost two times larger than those from two other hybrids. Undoubtedly, these estimations are semi-quantitative because the amplitude of ΔΨ depends on the efficiency of proteoliposome association with the collodion membrane in each experiment. (2) The increased ratio of the kinetic components in the microsecond time range to that in the millisecond time range of the photoelectric response of the ESR-Cherry and ESR-Trx incorporated in proteoliposomes corresponds to the decreased efficiency of transmembrane proton transport in comparison with the wild-type ESR from reverse reactions. Unlike two other hybrids, in Caf-ESR this ratio is similar to that in the wild type and even smaller. (3) The kinetics of light-induced changes of the transmembrane potential difference of ESR-Cherry and ESR-Trx at pH 7.5 exhibited large negative phases presumably associated with the opposite orientation of the protein in the proteoliposome bilayer. These negative phases were completely absent in Caf-ESR. Moreover, in the wild-type ESR, a small negative phase with τ ~1.3 ms was previously observed at pH 6.6, which was attributed to the presence of a small fraction of oppositely oriented protein in the membrane [33]. In Caf-ESR, it was absent even at pH 6.5. As a result, the kinetic constants of the electrogenic phases differ from those of the photocycle transitions in ESR-Cherry and ESR-Trx (see Appendix A), while in Caf-ESR, they are more similar (Appendix A).

The observed differences could be explained assuming the presence of the ESR-Cherry and ESR-Trx with opposite orientations in the proteoliposomes and superposition with opposite signs of the corresponding photovoltage responses through the electrometric study. In the kinetics of light-induced absorption changes, signals from differently oriented proteins in proteoliposomes have the same direction and their addition leads to significantly smaller differences in comparison with the kinetics of ΔΨ generation. This confirms the utility of the direct electrometry approach for the assessment of the orientation and functional state of the proton pumps in the lipid environment.

The obtained results indicate that fusion with Caf1M provides a highly unidirectional orientation of ESR in proteoliposomes with a fraction of correctly oriented molecules close to 100%. Ritzmann et al. obtained almost 100% opposite orientation of the PR molecules by placing fluorescent proteins at its N- or C-end [30]. In our work, the ESR fusions had mainly the same orientation as the wild-type protein independently of the fusion position. We presume that the obtained results could be explained by different procedures for liposome preparation used in these studies. In [30], the preformed liposomes were mixed with the purified protein that promoted preferential insertion of the molecules with the soluble domain outside the bilayer. The protocol described by Rigaud [21] does not include the initial preparation of the liposomes; instead, they are formed upon incubation with the protein solution. During this procedure, soluble domains have the possibility to be located both inside the lumen and in the external bulk; consequently, the direction of insertion is determined mainly by the membrane protein.

Intrinsic properties of the protein molecule (hydrophobicity, configuration, and charge) are responsible for obtaining its preferential orientation in proteoliposomes [21]. It was demonstrated earlier that the ESR molecule itself has mostly unidirectional orientation in the lipid bilayer [35]. We can conclude that its geometry/charge distribution promotes asymmetrical insertion in the N_out_-C_in_ direction. Presumably, C-terminal fusions with ESR also tend to preserve this orientation in the bilayer with their soluble domains (Trx or mCherry) inside the lumen. However, the limited internal volume of the proteoliposomes does not allow accommodation of all the fusion molecules in a single direction and, as a result, a fraction of the fusions acquires an opposite (N_in_-C_out_) orientation (Figure 6). The kinetics of the potential difference generation differs between two oppositely directed protein populations, resulting in the appearance of the negative phases that were detected in the ESR-Trx and ESR-Cherry photoelectric response. Notably, the relative contribution of the major negative phase in ESR-Trx (24%) was about two times lower than in ESR-Cherry (51%), possibly reflecting the decreased amount of the oppositely oriented ESR-Trx molecules due to the lower size of the thioredoxin fusion partner in comparison with mCherry.

On the contrary, the N-terminal position of Caf1M in the fusion does not create any steric constraints and further promotes the insertion of ESR in the N_out_-C_in_ direction (Figure 6). This is reflected in the absence of negative phases in the kinetics of ΔΨ generation in the Caf-ESR at neutral and mildly acidic pH values. We can speculate that fusion with Caf1M strengthens the natural ability of ESR to insert with its N-terminus facing the external bulk by complete prevention of insertion in the opposite direction.

## 4. Materials and Methods

### 4.1. Recombinant Gene Construction and Expression

To construct ESR-Cherry, ESR gene was amplified from pET-ESR plasmid [31] with primers T7prom and ESR_Bam ACATGGATCCGGACGTCAGCGTTTTTCCTT, digested with NdeI and BamHI, and cloned into pET32 plasmid together with mCherry coding sequence. Gene coding for mCherry was amplified using primers Bam-Cherry ATAAGGATCCGGTGGAGGTGGCTCTGTGAGCAAGGGCGAGGAG, and Cherry-Xho ACATCTCGAGCTTGTACAGCTCGTCCATGC and pmCherry-C1 (Clontech, Mountain View, CA, USA) as a template; and digested with BamHI and XhoI. Gene coding for *E. coli* thioredoxin (Trx) was amplified from pET32a with primers TrxBam TCATAGGATCCGGTGGAGGTGGCTCTAGCGATAAAATTATTCACCTGAC and TrxXho TCATACTCGAGGGCCAGGTTAGCGTCGAGG; and cloned into pESR-Cherry digested with BamHI and XhoI.

The coding sequence of Caf1M chaperone with its own signal sequence was obtained by PCR with primers Nde-Caf ACTAACATATGATTTTAAATAGATTAAGTACG and Caf-Nco GTTTGTATTCCAAAAATGTGACTTTAGGAGGTtccatggATAA from pCaf1M plasmid DNA [41]; and after digestion with NdeI and NcoI, cloned into pET32 plasmid together with ESR gene amplified with primers Nco-ESR ATAACCATGGGAGGTTCTGAAGAAGTCAATTTACTCGTTC and T7term, and digested with NcoI and XhoI.

pCherry-ESR was constructed by cloning the mCherry gene amplified with primers Pag_Cherry TACTATCATGAGTTCTGAAGATGTTATC and Cherry_Bam ACATGGATCCCTTGTACAGCTCGTCCATGC, together with the ESR gene amplified with Bam_ESR and ESR_Xho into the pET20b vector digested with NcoI and XhoI. All constructs were verified by sequencing (Evrogen, Moscow, Russia).

*E. coli* C41(DE3) cells were transformed with the resulting plasmids and grown in LB with ampicillin at 37 °C until OD at 560 nm reached 0.8. Expression of the recombinant genes was induced by addition of 0.2 mM IPTG. Incubation continued at 25 °C for 16 h in the presence of 5 µM all*-trans* retinal.

### 4.2. Hybrid Protein Purification

Harvested cells were resuspended in 50 mM Tris-HCl, pH 8.0, 5 mM EDTA, 20% sucrose with lysozyme (0.2 mg/mL) and disrupted by sonication. After centrifugation for 30 min at 6000× *g*, the obtained supernatant was ultracentrifuged for 1 h at 100,000× *g*. The resulting precipitate (total membrane fraction) was resuspended in 50 mM Tris-HCl, pH 8.0, and solubilized overnight by addition of 1% DDM. Supernatant after centrifugation at 20,000× *g* was applied onto Ni-Sepharose (GE Healthcare, Chicago, IL, USA) column, washed with buffer containing 30 mM Na-P, pH 7.4, 200 mM NaCl, 0.05% DDM, and 20 mM imidazole; and eluted in the same buffer containing 300 mM imidasole. The purified protein was concentrated and washed from imidasole using Ultracel YM-30 centrifugal filter devices (Merck Millipore, Burlington, MA, USA).

### 4.3. Protein Electrophoresis and Western Blot

Membrane proteins were separated by gel electrophoresis in 13% SDS-PAGE and transferred onto nitrocellulose membrane (Bio-Rad, Hercules, CA, USA). Bands were visualized using monoclonal antibodies to the hexahistidine tag conjugated with HRP (anti-His_6_, Invitrogen, Waltham, MA, USA).

### 4.4. Spectroscopic Characterization

Kinetics of flash-induced absorbance changes of the hybrid proteins were measured at characteristic wavelengths as described earlier for ESR [35]. Before measurements, the samples were diluted to achieve A_530_ = 0.1. Flashes (532 nm, 8 ns, 10 mJ) were from LS-2131M Nd-YAG Q-switched laser (LOTIS TII, Minsk, Belarus). Transient absorption changes were detected by photomultiplier and digitized by Octopus CompuScope 8327 (GaGe, Toronto, ON, Canada). Kinetic traces were fit with a sum of exponentials using Mathematica (Wolfram Research, Champaign, IL, USA). All experiments were repeated at least three times with the mean results presented.

### 4.5. Electrometric Time-Resolved Measurements of the Membrane Potential ΔΨ Generation

Reconstitution in proteoliposomes and photoelectrical measurements were performed as described in [35,36,37]. Liposomes were prepared from azolectin (20 mg/mL, Sigma, type IV-S, 40% *w*/*w* phosphatidylcholine content) by sonication at 22 kHz, 60 µA for 2 min in 1 mL of 25 mM HEPES-NaOH buffer, pH 7.5. Reconstitution of ESR into proteoliposomes was performed by mixing the liposomes with ESR in 1.5% (*w*/*v*) OG at the lipid/protein ratio of 100:1 (*w*/*w*) for 30 min in the dark. The detergent was removed according to [21]. Bio-Beads SM-2 (Bio-Rad) was added to the mixture in a 20-fold excess (by weight) and suspension was stirred for 3 h at room temperature. The proteoliposomes were separated from absorbent by decanting and pelleted by centrifugation at 140,000× *g* at 4 °C for 1 h in a Beckman L-90K ultracentrifuge. The pellet was resuspended in 25 mM HEPES-NaOH buffer (pH 7.5). A home-made electrometric setup with 100 ns time resolution was used in combination with a pulsed Nd-YAG laser (532 nm, 12 ns, 40 mJ). Flash-induced kinetics of ΔΨ generation was measured by Ag+/AgCl electrodes at different sides of the membrane. For measurements at different pH values, equivalent (~25 mM) buffer solutions were used: MES, HEPES, Tris, or CHES. All experiments were repeated at least three times.

### 4.6. Data Analysis

The data from time resolved optical and photoelectric measurements were fit into a sum of individual exponents using Pluk [59], Origin (OriginLab Corporation, Northampton, MA, USA), and MATLAB (The Mathworks, South Natick, MA, USA) software, as described earlier for the wild-type and mutants of ESR, BR, cytochrome oxidase, and other proteins [43,60,61,62,63,64,65,66].

## 5. Conclusions

In conclusion, in this paper we demonstrate the utility of N-terminal fusion with Caf1M protein for highly efficient unidirectional insertion of ESR into proteoliposomes. The developed approach could be useful for obtaining oriented incorporation of other membrane proteins into artificial membranes for their functional studies and biotechnological applications.

## Figures and Tables

**Figure 1 ijms-24-07369-f001:**
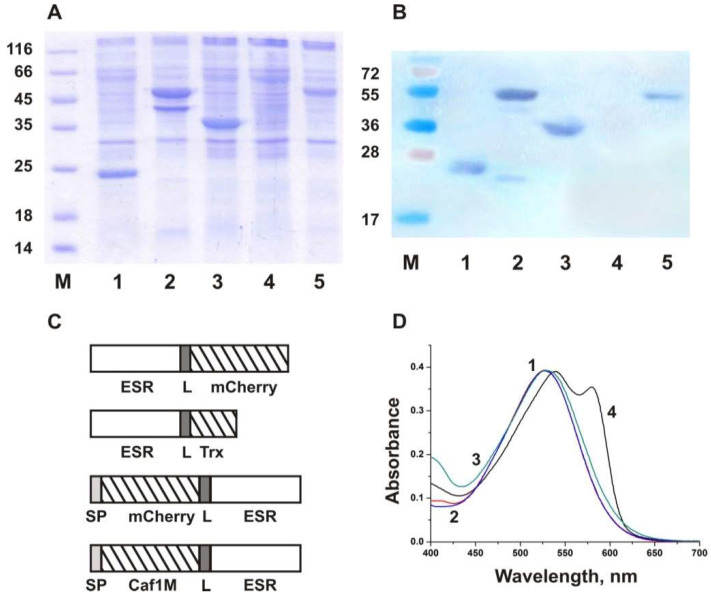
Expression of the hybrid proteins. (**A**) SDS-PAGE and (**B**) Western blot using anti-His antibodies of the membrane fractions of *E. coli* C41(DE3) cells expressing ESR (lane 1), ESR-Cherry (lane 2), ESR-Trx (lane 3), Cherry-ESR (lane 4), and Caf-ESR (lane 5). M—protein molecular weight markers (Thermo Scientific, Waltham, MA, USA). (**C**) Structural schemes of the hybrid proteins. SP, signal peptide (PelB for mCherry, a native one for Caf1M); L, linker. (**D**) Absorption spectra at pH 7.0 of the purified wild type ESR (1), and the hybrid proteins ESR-Trx (2), Caf-ESR (3), and ESR-Cherry (4).

**Figure 2 ijms-24-07369-f002:**
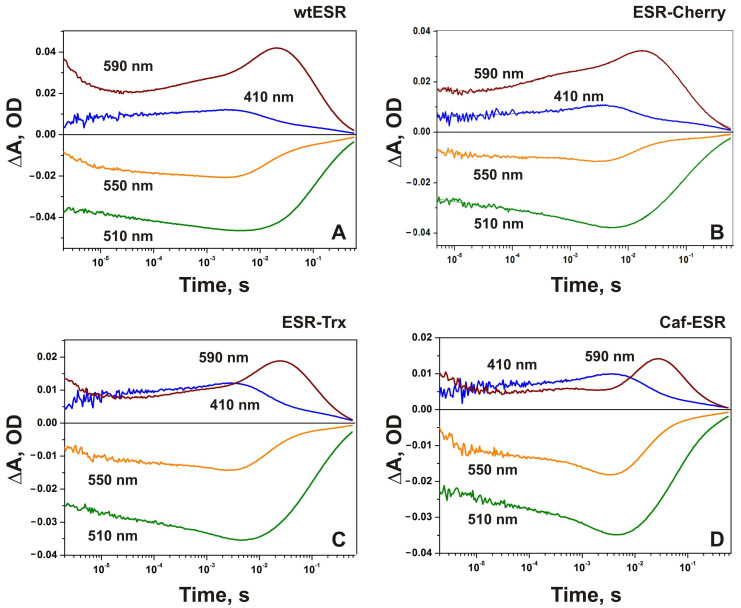
Light-induced absorption changes at four characteristic wavelengths (410 nm, 590 nm, 550 nm, and 510 nm) at pH 9.0 in DDM micelles containing: (**A**) wild-type ESR; (**B**) ESR-Cherry; (**C**) ESR-Trx; (**D**) Caf-ESR.

**Figure 3 ijms-24-07369-f003:**
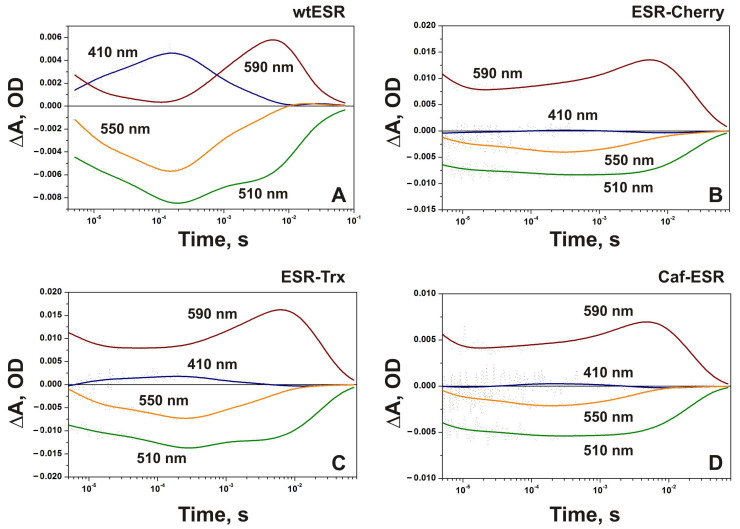
Light-induced absorption changes at four characteristic wavelengths (410 nm, 590 nm, 550 nm, and 510 nm) at pH 7.5 in proteoliposome suspensions containing: (**A**) wild-type ESR; (**B**) ESR-Cherry; (**C**) ESR-Trx; (**D**) Caf-ESR.

**Figure 4 ijms-24-07369-f004:**
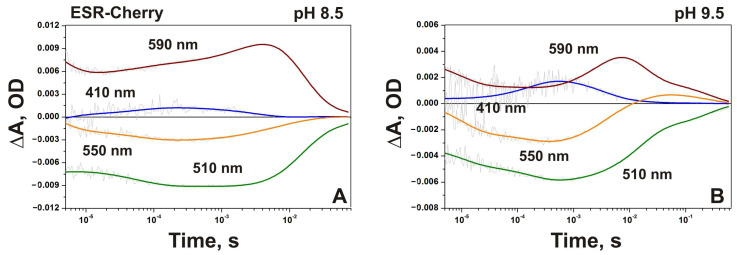
Light-induced absorption changes at four characteristic wavelengths (410 nm, 590 nm, 550 nm, and 510 nm) at pH 8.5 (**A**) and 9.5 (**B**) in proteoliposome suspensions containing ESR-Cherry.

**Figure 5 ijms-24-07369-f005:**
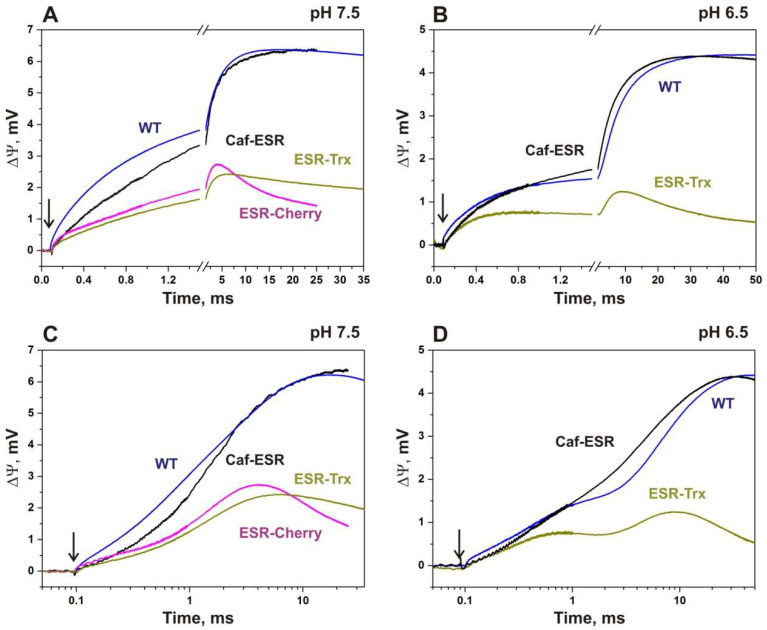
Kinetics of light-induced changes in membrane potential difference, ΔΨ, in proteoliposomes containing wild-type ESR and hybrid proteins at pH 7.5 (**A**,**C**) and 6.5 (**B**,**D**). Photoelectric traces are shown on a piecewise linear (**A**,**B**) and logarithmic (**C**,**D**) time scale. The trace of the wild-type ESR is normalized by the maximum amplitude with that of the Caf-ESR. For the ESR-Cherry, only the trace of photoelectric response at pH 7.5 is shown. The time point of the flash is indicated by an arrow.

**Figure 6 ijms-24-07369-f006:**
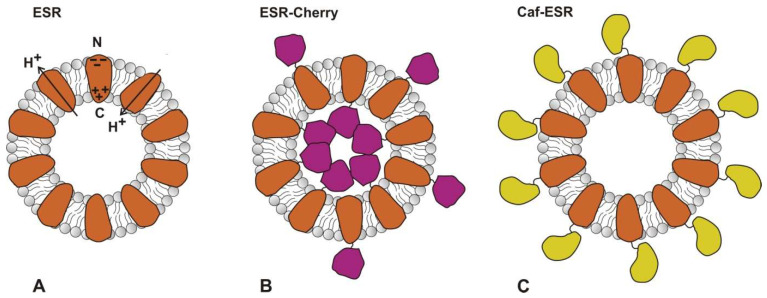
Schematic drawings of proteoliposomes containing ESR and hybrid proteins. (**A**) The wild type ESR is mainly oriented in the N_out_-C_in_ direction. Only a small portion of molecules (about 5%) possesses an opposite orientation. (**B**) C-terminal soluble domain of ESR-Cherry points to the lumen; however, due to its limited volume, some fraction of molecules inserts in the N_in_-C_out_ direction. ESR-Trx presumably inserts by the same mechanism. (**C**) N-terminal position of the soluble domain in Caf-ESR promotes unidirectional N_out_-C_in_ orientation.

**Table 1 ijms-24-07369-t001:** Properties of the fusion partners used in the study.

Protein	Molecular Weight, kDa	pI **	Reference
mCherry	26.6	5.63	[39]
Trx	11.7	4.67	[40]
Caf1M *	26.8	7.9	[41]

* Data for the mature protein are provided. ** Calculated value.

**Table 2 ijms-24-07369-t002:** Time constants (ms) of the photocycle in wild-type ESR and the hybrid proteins in DDM micelles at pH 9.0.

Protein	τ1	τ2	τ3	τ4	τ5	τ6
wt ESR	0.0044 ± 0.0002	0.13 ± 0.008	2.24 ± 0.13	12.1 ± 0.38	82.6 ± 3.6	310 ± 14
ESR-Cherry	0.0092 ± 0.0011	0.18 ± 0.013	2.9 ± 0.27	10.0 ± 0.77	73.6 ± 5.2	293 ± 21
ESR-Trx	0.008 ± 0.0006	0.15 ± 0.02	2.28 ± 0.14	13.3 ± 0.52	79.6 ± 5.4	319 ± 20
Caf-ESR	0.004 ± 0.0003	0.08 ± 0.01	2.23 ± 0.11	14.2 ± 0.66	60.0 ± 3.9	322 ± 29

## Data Availability

The data are available from the corresponding author on reasonable request.

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
