# Peer review of "Oriented Insertion of ESR-Containing Hybrid Proteins in Proteoliposomes"

_ijms, 2023, doi:10.3390/ijms24087369_

Round 1

Reviewer 1 Report

In this article Petrovskaya et al attempt to develop a method to obtain unidirectionality in orientation when reconstituting purified membrane protein from detergent micelles into proteoliposomes. They do this by creating hybrids where small soluble proteins are added to the N or C termini of the membrane protein ESR. The title and abstract are exciting, as this is a problem for many people working with purified, reconstituted proteins. However, whilst well carried out technically, the findings of the paper are underwhelming. The example protein used (ESR) naturally incorporates into proteoliposomes with 95% facing the same way. Addition of an N-terminal Caf1M protein does increase the orientation preference from 95% to 100%, but this is a relatively small effect. It would be much more impressive/useful in the field if they used an example protein which on its own shows something like 40-60% in the same direction. Other hybrid proteins, with mCherry or thioredoxin added to the C-terminus of ESR did not show any improvements in orientation, and affected the function of the membrane protein.

Major issues:

To be of wide interest to membrane protein researchers I think they really need to show the addition of Caf1M can improve the unidirectionality of a protein which doesn’t show 95% the same orientation on its own. Whilst focussing just on ESR, the impact of this work will be very limited.

Page 15 Discussion. The authors discuss differences in reconstitution method which may have an impact on the preferential insertion. It would be interesting if they tried both methods of reconstitution to compare.

Minor points:

A schematic of the photocycle of ESR, showing the different states (K, M, N etc) and their absorbances would make the paper more accessible to those not familiar with ESR.

There are several typos, grammatical errors and different font sizes throughout the paper.

Author Response

Major issues:

To be of wide interest to membrane protein researchers I think they really need to show the addition of Caf1M can improve the unidirectionality of a protein which doesn’t show 95% the same orientation on its own. Whilst focussing just on ESR, the impact of this work will be very limited.

> We are grateful to the Reviewer for the positive evaluation of our manuscript. Definitely, the ultimate goal of our work was to achieve unidirectional orientation of a target protein which is intrinsically problematic to obtain. In the current paper, ESR was used as a model protein which was extensively studied previously. The available information about its properties was used to access the impact of the fusion partners on its function. Furthermore, we were inspired by the studies of some ESR mutants which demonstrated only small electrogenic responses in our experiments, presumably, as a result of mixed orientation in proteoliposomes. In the future work, we will apply the developed approach to obtain data with such mutants and other proteins.

Page 15 Discussion. The authors discuss differences in reconstitution method which may have an impact on the preferential insertion. It would be interesting if they tried both methods of reconstitution to compare.

> The protocol for proteoliposomes preparation for direct electrometry measurements is based on the described reconstitution procedure (Rigaud et al., 1995). We agree that comparison of different methods is an interesting task; however, it was not the focus of the current study.

Minor points:

A schematic of the photocycle of ESR, showing the different states (K, M, N etc) and their absorbances would make the paper more accessible to those not familiar with ESR.

>We included the proposed scheme into the text (p. 5).

There are several typos, grammatical errors and different font sizes throughout the paper.

>We apologize for the mistakes. The manuscript was thoroughly edited.

Reviewer 2 Report

The work is thorough and sound. Experiments are nicely designed, data are well-presented. 

Author Response

We are grateful to the Reviewer for the positive evaluation of our manuscript.

Reviewer 3 Report

Review report

Journal: IJMS (ISSN 1422-0067)

Manuscript ID: ijms-2317103

Type: Article

Title: “Oriented insertion of ESR-containing hybrid proteins in lipid membranes”.

Authors: Lada E. Petrovskaya, Evgeniy P. Lukashev, Mahir D. Mamedov, Elena A. Kryukova, Sergei P. Balashov, Dmitry A. Dolgikh, Andrey B. Rubin, Mikhail P. Kirpichnikov, Sergey A. Siletsky *

Section: Molecular Biology

Special Issue: Ion Pumps: Molecular Mechanisms, Structure, Physiology

General.

This work focuses on rhodopsins. The authors state that to study their properties in a native-like environment, these proteins are incorporated into proteoliposomes. However, unidirectional protein orientation in artificial membranes is rarely observed.

In this study, the authors aimed to obtain proteoliposomes with unidirectional orientation using a retinal proton pump protein from Exiguobacterium sibiricum (ESR) as a model. The authors obtained and characterised three ESR hybrids with soluble protein domains (mCherry or thioredoxin at the C-terminus and Caf1M chaperone at the N-terminus).

According to the results, the photocycle of the hybrid proteins incorporated into proteoliposomes showed a higher pKa of the M-state accumulation compared to that of the wild-type ESR. ESR-Cherry and ESR-Trx showed large negative electrogenic phases and an increase in the relative amplitude of the kinetic components in the microsecond time range in the kinetics of membrane potential generation, indicating a decrease in the efficiency of transmembrane proton transport.

On the other hand, Caf-ESR showed native-like kinetics of membrane potential generation and the corresponding electrogenic stages. The experiments showed that the hybrid with Caf1M promotes the unidirectional orientation of the ESR in proteoliposomes.

Major concerns

1. Overall, the experiments appear to be well designed and performed. However, the measurements are not statistically summarised. The authors should provide evidence of statistical analysis or whether the experiments are a single experimental measurement.

2. The origin of the orientation of many membrane proteins of most eukaryotic cells is still far from being understood, even those present in the membranes of the oldest living microorganisms. In this sense, what is the significance of unidirectionality in the study systems considered in this work, and is it just an attempt to mimic some natural membranes? This should be made clear in a few lines of the introduction.

3. In line 65, the authors should at least mention some reference: Is it always necessary for proteins to be oriented in only one direction?

4. In the "Methodology" section, the authors do not mention the composition of the proteoliposomes at all. They only refer to three papers (line 151), but do not mention the lipid composition of the liposomes or the method used to prepare them. They make no mention of the z-potential (charge of the liposomes), no mention of the presence or amount of charged lipids, the size of the liposomes, the nature of the internal/external solution, etc. These experimental details are crucial (perhaps too important to be overlooked) as the orientation of membrane proteins or other types of molecules is largely moderated by the nature of the lipid bilayer of proteoliposomes. This should have been mentioned in detail and, as a control, the interactions between ESR, ESR-Cherry and ESR-Caf should have been detailed. What was the reason for this to be omitted?

5. In line 323 it is mentioned "...due to the instability of ESR-Cherry containing proteoliposomes at pH 6.5", the authors should add a reference to support this sentence. Is this an experimental finding of the authors?

6. In lines 465-467 the authors should add a reference to the quantitative relationship between the "accelerated passive discharge" of liposomes and the "permeability" of liposomes. Does this permeability refer to a particular type of molecule? Please specify.

7. What type of phospholipids are you referring to in lines 522-523? It is not possible to generalise in this sentence and it must be very specific. The same applies to line 538: What type of polar headgroup? Please specify.

8. On line 616 the authors mention: "the limited internal volume of the proteoliosomes", What is the value of the internal volume? How were the liposomes obtained and what was their size distribution?

9. Considering all the above comments, I think the authors should modify the title as "lipid membranes" is a very broad term and the authors have not even specified the type of membrane or phospholipids used.

Minor concerns

10. Lines 56-63 have different font sizes.

11. Lines 339-343 have a different font size from the rest of the text.

12. Reference 43 is underlined in the text.

13. Figure 6 should illustrate (schematically) the charges of the inserted proteins, just to get an overall idea.

Overall, I consider that the article could have significant potential and eventually provide a valuable contribution to the field. However, major corrections are required before it can be accepted for publication. If the problems identified in this review are addressed, I am confident that the article will be a valuable contribution to the field and deserves to be accepted for publication.

Author Response

Major concerns

  1. Overall, the experiments appear to be well designed and performed. However, the measurements are not statistically summarised. The authors should provide evidence of statistical analysis or whether the experiments are a single experimental measurement.

>We are grateful to the Reviewer for the positive evaluation of our work. All experiments were repeated at least three times. This information has been included into Materials and Methods section.

  1. The origin of the orientation of many membrane proteins of most eukaryotic cells is still far from being understood, even those present in the membranes of the oldest living microorganisms. In this sense, what is the significance of unidirectionality in the study systems considered in this work, and is it just an attempt to mimic some natural membranes? This should be made clear in a few lines of the introduction.

>As we stated (line 69), a mixture of orientations results in a decreased functional signal. Now we have specified that this implies impaired ion transport properties when speaking about ion transport rhodopsins. It was also mentioned that this can limit their studies and application. Corresponding text has been added to Introduction (p.2).

  1. In line 65, the authors should at least mention some reference: Is it always necessary for proteins to be oriented in only one direction?

>As we mentioned (line 64), the uniform protein orientation in the membrane is a highly desirable situation which is beneficial for their studies and application. It is especially essential for proteins involved in unidirectional electrogenic ion transport. We have added a couple of references in support of this statement (refs 20, 21).

  1. In the "Methodology" section, the authors do not mention the composition of the proteoliposomes at all. They only refer to three papers (line 151), but do not mention the lipid composition of the liposomes or the method used to prepare them. They make no mention of the z-potential (charge of the liposomes), no mention of the presence or amount of charged lipids, the size of the liposomes, the nature of the internal/external solution, etc. These experimental details are crucial (perhaps too important to be overlooked) as the orientation of membrane proteins or other types of molecules is largely moderated by the nature of the lipid bilayer of proteoliposomes. This should have been mentioned in detail and, as a control, the interactions between ESR, ESR-Cherry and ESR-Caf should have been detailed. What was the reason for this to be omitted?

>We apologize for the inconvenience. The protocol for preparation of the proteoliposomes was added to the Materials and Methods section. It is a standard method described by Rigaud in 80-90s, so the characteristics of the produced liposomes were described previously. We added a link to the comprehensive review (Rigaut 1995 BBA and references therein) in the text. We also mentioned the composition of the lipid used (azolectin) in the Discussion (p. 13).

  1. In line 323 it is mentioned "...due to the instability of ESR-Cherry containing proteoliposomes at pH 6.5", the authors should add a reference to support this sentence. Is this an experimental finding of the authors?

>Yes, it was an experimental finding revealed in the current study.

  1. In lines 465-467 the authors should add a reference to the quantitative relationship between the "accelerated passive discharge" of liposomes and the "permeability" of liposomes. Does this permeability refer to a particular type of molecule? Please specify.

>The appearance of fast membrane discharge components indicates an increase in the permeability of liposomes for protons similar to the addition of uncouplers (Drachev 1981). Presumably, the proton leakage occurs either through the protein-lipid contact or through the protein itself (for example, as a result of reverse reactions in a photocycle). We have added a corresponding clarification in the text (p. 9).

  1. What type of phospholipids are you referring to in lines 522-523? It is not possible to generalise in this sentence and it must be very specific. The same applies to line 538: What type of polar headgroup? Please specify.

>We specified the type of lipids in the mentioned paper (p. 12). Azolectin which was used for proteoliposome preparation contains phosphatidylcholine, phosphatidylethanolamine, phosphatidylinositol and other minor lipids. We added this information in the Discussion (p. 13).

  1. On line 616 the authors mention: "the limited internal volume of the proteoliosomes", What is the value of the internal volume? How were the liposomes obtained and what was their size distribution?

> We added a description of the liposome preparation technique to the Materials and Methods section. The size of liposomes obtained in the used conditions is about 200 nm (Rigaud 1995). The internal volume of the liposomes can be calculated from this value; however; in the Discussion we used this estimation in qualitative sense, in contrast with the bulk solution where the space for the protein molecules is not limited.

  1. Considering all the above comments, I think the authors should modify the title as "lipid membranes" is a very broad term and the authors have not even specified the type of membrane or phospholipids used.

>The title was changed to Oriented Insertion of ESR-Containing Hybrid Proteins in Proteoliposomes.

Minor concerns

  1. Lines 56-63 have different font sizes.
  2. Lines 339-343 have a different font size from the rest of the text.
  3. Reference 43 is underlined in the text.

>We apologize for the mistakes. The manuscript was thoroughly edited.

  1. Figure 6 should illustrate (schematically) the charges of the inserted proteins, just to get an overall idea.

>We have modified Fig. 6 accordingly.

Round 2

Reviewer 1 Report

I still think the interest and impact of this work will be very limited, as it simply shows improvement of a process from 95% to 100%, for one particular protein. However, the work is well carried out and I have no issues with the data or interpretation. I think the few changes made do offer small improvements. I understand that the more substantial changes suggested would be a lot of work.

Reviewer 3 Report

Dears authors,

Thank you for submitting the revised version of your manuscript to IJMS, and for taking into account my suggestions during the review process.

I have carefully reviewed your revised manuscript and I consider that all the points I raised have been adequately addressed. The corrections made have improved the clarity and accuracy of the manuscript.

Based on my assessment, I strongly recommend that this version of the manuscript be accepted for publication in IJMS.
